# Anonymization for Skeleton Action Recognition

**Saemi Moon**[1]* **Myeonghyeon Kim**[2]* **Zhenyue Qin**[3] **Yang Liu**[3] **Dongwoo Kim**[1]

**POSTECH**[1]   **Scatter Lab**[2]   **Australian National University**[3]

## Abstract

Skeleton-based action recognition attracts practitioners and researchers due to the lightweight, compact nature of datasets. Compared with RGB-video-based action recognition, skeleton-based action recognition is a safer way to protect the privacy of subjects while having competitive recognition performance. However, due to improvements in skeleton recognition algorithms as well as motion and depth sensors, more details of motion characteristics can be preserved in the skeleton dataset, leading to potential privacy leakage. We first train classifiers to categorize private information from skeleton trajectories to investigate the potential privacy leakage from skeleton datasets. Our preliminary experiments show that the gender classifier achieves 87% accuracy on average, and the re-identification classifier achieves 80% accuracy on average with three baseline models: Shift-GCN, MS-G3D, and 2s-AGCN. We propose an anonymization framework based on adversarial learning to protect potential privacy leakage from the skeleton dataset. Experimental results show that an anonymized dataset can reduce the risk of privacy leakage while having marginal effects on action recognition performance even with simple anonymizer architectures.

## 1 Introduction

Action recognition has been widely studied in many applications such as sports analysis (Tran et al., 2018), human-robot interaction (Fanello et al., 2013), and intelligent healthcare services (Saggese et al., 2019). To employ the recognition system appropriately, one must ensure that private information is not abused before and after analysis. Skeleton-based action recognition can be alternative to video-based recognition. Due to the advance in depth and motion sensors, details of motion characteristics can be preserved in the skeleton dataset. Compared with RGB videos, the skeleton dataset seems to expose fewer details on participants. It is often challenging to identify sensitive information such as gender or age from a skeleton to compare with the RGB video to the naked eye.

We raise a question about the privacy-safeness of skeleton datasets. To check potential privacy leakage from skeletons, we conduct experiments on identifying gender or identity with Shift-GCN (Cheng et al., 2020), MS-G3D (Liu et al., 2020), and 2s-AGCN (Shi et al., 2019). Based on our analysis, a properly trained classifier can predict the private information accurately. Therefore, the skeletons are not safe from the privacy leakage problem.

This work aims to develop a framework that can anonymize skeleton datasets while preserving critical action features for recognition. To this end, we propose a minimax framework to anonymize the skeletons. With RGB-video datasets, object detection followed by blurring or inpainting with pre-trained generative models is often employed to anonymize datasets (Yang et al., 2021; Hukkelås et al., 2019). However, these methods cannot be directly applied to the skeleton dataset.

---

*Equal contribution

2022 Trustworthy and Socially Responsible Machine Learning (TSRML 2022) co-located with NeurIPS 2022.

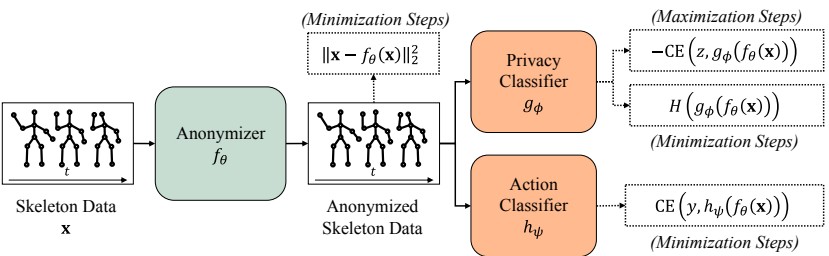

Figure 1: Anonymization framework. The framework consists of three sub-networks: 1) anonymizer $f_\theta$, 2) privacy classifier $g_\phi$, and 3) action classifier $h_\psi$. The dashed box represents the losses used in minimization and maximization steps with adversarial learning. Note that the privacy classifier uses a separate loss for minimization and maximization in setup. The parameter of the action classifier $\psi$ is pre-trained and not updated during anonymizer training.

The minimax framework consists of an anonymizer network with two sub-networks designed to predict action and private information. The anonymizer removes private information from skeletons, and then the output skeleton is fed into action and privacy classifiers separately. We maximize the accuracy of the action classifier while minimizing the identifiability of private information with the other classifier. In addition, we enforce the anonymized skeleton similar to the original one to make sure they are visually indistinguishable from each other. Experimental results show that the proposed algorithm results in an effective anonymizer.

We summarize our contributions as follows:

- We empirically show potential privacy leakage from widely-used skeleton datasets such as NTU60 (Shahroudy et al., 2016) and ETRI-activity3D(Jang et al., 2020).

- We develop a skeleton anonymization network based on action and sensitive variable classifiers.

- We propose a learning algorithm based on the adversarial learning method to anonymize skeletons.

- We show that the anonymized skeletons are more robust to privacy leakage while still enjoying high action recognition accuracy.

## 2   Skeleton Anonymization

In this section, we propose a framework for the skeleton anonymization model.

**Anonymization framework.**    Let $\vec{x} \in \mathbb{R}^{T \times D \times 3}$ be 3D coordinates of $D$ joints over $T$ frames, and $y \in \mathcal{Y}$ be an action label for a given skeleton sequence $\vec{x}$, where $\mathcal{Y}$ is a set of actions to be recognized. Let $z \in \mathcal{Z}$ be private information related to the skeleton sequence $\vec{x}$, e.g., gender or identity, where $\mathcal{Z}$ is a set of possible private labels.

We aim to develop an anonymization network that can effectively remove private information from skeleton datasets while maintaining the recognizability of actions from the anonymized skeletons. To do this, we propose a minimax framework consisting of three different neural network components. Let $f_\theta : \mathbb{R}^{T \times D \times 3} \to \mathbb{R}^{T \times D \times 3}$ be an anonymizer network aiming to remove sensitive information from the input skeletons, $h_\psi : \mathbb{R}^{T \times D \times 3} \to \mathcal{Y}$ be an action classifier, and $g_\phi : \mathbb{R}^{T \times D \times 3} \to \mathcal{Z}$ be a privacy classifier that predicts sensitive personal information. Our goal is to train an anonymizer $f_\theta$ whose output can maximally confuse the classification performance on the private variables. On the other hand, the output of the anonymizer should keep all relevant information for recognizing action to preserve the performance of the action classifier $h_\psi$. In other words, the output should not be very different from the original skeletons since the anonymized skeletons can be recognizable by naked eyes. To satisfy all requirements, we formalize the anonymization via the following minimax objective:

$$\min_{\theta} \max_{\phi} \mathbb{E} \left[ \text{CE}\left(y, h_\psi\left(f_\theta(\vec{x})\right)\right) - \alpha \, \text{CE}\left(z, g_\phi\left(f_\theta(\vec{x})\right)\right) + \beta ||\vec{x} - f_\theta(\vec{x})||_2^2 \right], \qquad (1)$$

where CE is the cross entropy, and $\alpha$ and $\beta$ are hyperparameters controlling the importance of the privacy classification and the reconstruction error, respectively. The reconstruction error between the original and anonymized skeleton data $||\vec{x} - f_\theta(\vec{x})||_2^2$ ensures the anonymized skeletons are similar to the original ones. To maximize the objective, the private classifier needs to classify the private label $z$ correctly. To minimize the objective, the anonymizer makes the actions easily identifiable by action classifier $h_\psi$ while making the private classifier misclassify the private label $z$. To simplify the learning process, we use a pre-trained action classifier and fix the parameters of the action classifier during training. The fixed action classifier constrains the anonymized skeleton compatible with the pre-trained model. The anonymized skeletons are also likely to work well with other pre-trained classifiers available.

Minimizing the objective w.r.t $\theta$ can make the anonymizer fool the private classifier. However, one may exploit this fact to infer the true label. For example, in a binary classification problem, the true label can be obtained by choosing the opposite of the prediction. To avoid this issue, we minimize the entropy of classified outputs during the minimization step:

$$\min_{\theta} \mathcal{L}_{\text{adv}} = \min_{\theta} \mathbb{E} \left[ \text{CE}\left(y, h_\psi\left(f_\theta(\vec{x})\right)\right) - \alpha H\left(g_\phi\left(f_\theta(\vec{x})\right)\right) + \beta ||\vec{x} - f_\theta(\vec{x})||_2^2 \right], \qquad (2)$$

where $H\left(g_\phi\left(f_\theta(\vec{x})\right)\right)$ is the entropy of the distribution of private labels predicted from the anonymized skeleton. Therefore, the optimal anonymizer yields the most confusing skeletons to the private classifier. In the maximization step, we still maximize the negative cross entropy $-\alpha \, \text{CE}\left(z, g_\phi\left(f_\theta(\vec{x})\right)\right)$ w.r.t. $\phi$ to train the private classifier. Figure 1 shows the overall framework for the data anonymization.

Alternating minimization and maximization are often employed to solve a minimax objective as shown in the generative adversarial network Goodfellow et al. (2014). Following previous work, we also use the alternating algorithm to optimize the objective. In this work, the adversarial learning algorithm starts with pre-trained classifiers $g_\phi$ and $h_\psi$ to make the learning stable.

**Anonymizer networks.** The anonymizer $f_\theta$ can be any prediction model that modifies skeletons while preserving the original dimension. We employ two simple neural network architectures for the anonymizer: 1) Residual networks (He et al., 2016) and 2) U-net architectures (Ronneberger et al., 2015). The detailed each anonymizer network architecture is provided in Appendix A.1.

## 3 Experiments

In this section, we demonstrate the performance of the proposed framework for anonymizing skeleton datasets. We use two publicly available datasets: ETRI-activity3D (Jang et al., 2020) and NTU RGB+D 60 (NTU60) (Shahroudy et al., 2016). For the ETRI-activity3D dataset, we anonymize the gender information from the skeletons. For the NTU60 dataset, we anonymize the identity of the skeletons. The detailed information and experimental setups for these datasets are provided in Appendix A.2.

### 3.1 Privacy Leakage

To verify privacy leakage from each dataset, we first check the performance of gender classification and re-identification task. To train gender classifier and re-identification task, three popular baseline models, Shift-GCN (Cheng et al., 2020), MS-G3D (Liu et al., 2020), and 2s-AGCN (Shi et al., 2019), are adopted. For gender classifier with MS-G3D, we use MS-G3D without a G3D module. This makes training faster without losing too much accuracy. We train multiple times for gender classifier and re-identification task. Each model is trained with a different random initialization.

After training, the gender classifier achieves 87% accuracy on average. The re-identification task achieves 80% and 97% for top-1 and top-5 accuracy, respectively. The detailed results are available in Appendix B. As the results suggest, the privacy information can be easily predicted by a classification

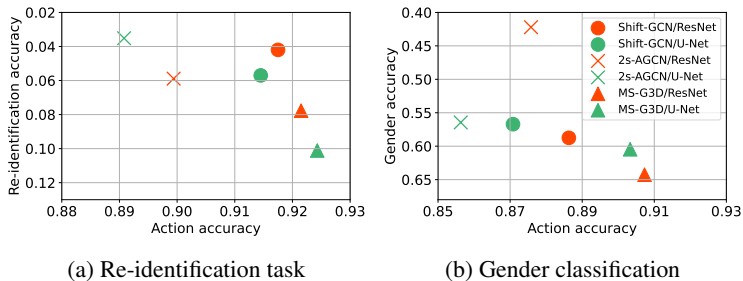

(a) Re-identification task  (b) Gender classification

Figure 2: Action and privacy accuracy of three baseline models with two different anonymizers after anonymization. $y$-axis is reversed. Note that, before anonymization, the average top-1 re-identification accuracy is 80%, and the average gender classification accuracy is 87%.

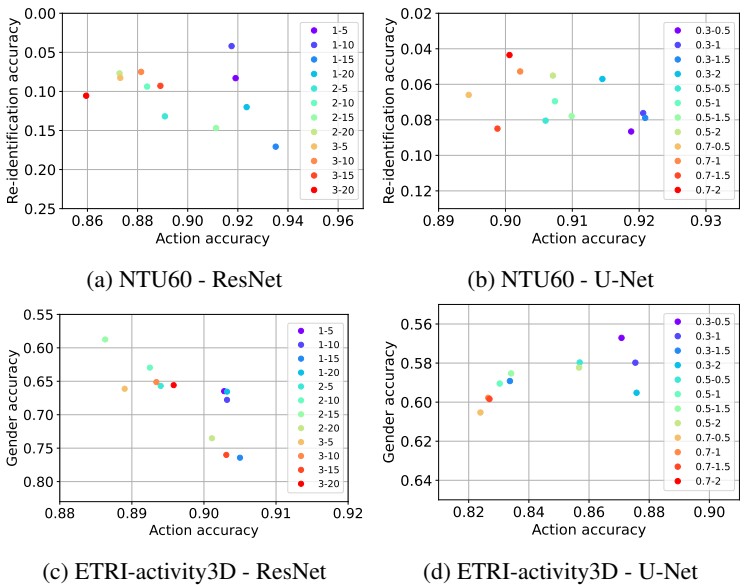

(a) NTU60 - ResNet  (b) NTU60 - U-Net

(c) ETRI-activity3D - ResNet  (d) ETRI-activity3D - U-Net

Figure 3: The trade-off between action accuracy and privacy accuracy based on a different configuration of hyperparameter $\alpha$ and $\beta$ on NTU60 and ETRI-activity3D with two anonymizer networks (legend: $\alpha$-$\beta$). Note that the y-axis is reversed.

model trained with private labels. Note that the test splits do not contain the subject used for gender classification. This reveals the generalizability of gender classification to the unseen subjects. Also, for the re-identification task, train split and test split have different camera IDs, so the same person appears in both sets with different views. This result indicates that the joint trajectory contains personal traits that can be easily exploited to identify a person.

## 3.2 Anonymization Results

Our preliminary study indicates that gender and identification can leak enough from training. Based on the results obtained in the previous experiments, we evaluate the performance of anonymization with an adversarial learning algorithm. For each task, we use two pre-trained classifiers for action and privacy, respectively. One classifier is used to initialize the adversarial algorithm, and the other is used to measure the accuracy after anonymization.

Figure 2 shows the results of anonymization with re-identification task and gender classification. In general, we observe that we can dramatically decrease the privacy accuracy while minimally sacrificing the action recognition accuracy. We also observe more leakage of private information when the action accuracy is relatively higher. Note that we use a balanced test set for identity anonymization.

Table 1: Comparison between our method and other alternative approaches. This experiment is conducted with the residual anonymizer on NTU60.

| Method | | Action. | Iden. |
|---|---|---|---|
| Not-anonymized | | 0.9510 | 0.8095 |
| Random noise | $\sigma = 0.001$ | 0.7565 | 0.7450 |
| | $\sigma = 0.005$ | 0.4430 | 0.3240 |
| | $\sigma = 0.010$ | 0.2660 | 0.1735 |
| | $\sigma = 0.020$ | 0.1265 | 0.1020 |
| | $\sigma = 0.050$ | 0.0455 | 0.0840 |
| | $\sigma = 0.100$ | 0.0450 | 0.0715 |
| Adversarial attack | Attacked | 0.9435 | 0.0000 |
| | Non-Attacked | 0.9435 | 0.3621 |
| Our method | | 0.9175 | 0.0420 |

Figure 4: Five frames of the original (top) and the gender anonymized (bottom) skeletons for an action "wiping face with a towel" from ETRI-activity3D. The subject is an elderly female.

Since NTU60 has 40 subjects, one can achieve 2.5% accuracy with random classification. For the gender classification, one can achieve 50% accuracy with a random classifier on the test set.

One would expect the trade-off between action accuracy and privacy accuracy based on the choice of hyperparameters $\alpha$ and $\beta$. The choice of the best anonymization model may vary depending on the application. In this work, we report the performance of the best model based on *action accuracy × (1 - re-identification accuracy)* from the results of various configurations. According to our metric, ResNet ($\alpha$:1, $\beta$:10) and U-Net ($\alpha$:0.3, $\beta$:2) models are chosen as representative model. Note that one may use a different metric to select a model given different application scenarios. We use the best configuration obtained from the baseline model, Shift-GCN, to train the other baseline models.

**Trade-off Analysis.** We vary the value of $\alpha$ and $\beta$ to check the trade-off between action accuracy and privacy leakage based on different configurations of hyperparameters. We use Shift-GCN as a baseline model for analysis. Figure 3a and Figure 3b show the result with various hyperparameter configurations on the identity anonymization task. Figure 3c and Figure 3d show the result of the gender anonymization task. We can observe that given a fixed $\alpha$, increasing $\beta$ increases the chance of privacy leakage as well as the action accuracy showing the presence of the trade-off between the action accuracy and privacy leakage.

**Comparison with Alternative Approaches.** Since we propose privacy leakage for the first time, no anonymization method removes privacy information while remaining action accuracy high. Therefore, we consider two alternative approaches to anonymize privacy information by modifying skeleton data potentially. (1) Random noise: As a baseline, we randomly inject white noise drawn from the zero mean normal distributions with varying variances to the original skeleton. (2) Adversarial attack method: Adversarial attack is a technique that makes a model fool by perturbing input data. There are several adversarial attack research on skeleton action recognition (Liu et al., 2020; Wang et al., 2021; Diao et al., 2021). We use Wang et al. (2021) method to attack privacy information.

Table 1 shows the results of comparing our method to other approaches. The random noise cannot preserve action information while reducing privacy leakage. The results with an adversarial attack

show that attacked skeleton data success in removing privacy information to the target model, i.e., the identification accuracy of the attacked model is zero. However, identification accuracy remains relatively high for the other pre-trained model, which has not been attacked. The adversarial attack-based anonymization is model-specific and difficult to generalize to the unseen models, whereas anonymized skeleton data with our proposed framework performs relatively well with any pre-trained model.

**Qualitative Analysis.** To qualitatively understand the effect of anonymization, we visualize one example from the ETRI-activity3D dataset before and after anonymization in Figure 4. The top and bottom rows show five selected frames before and after anonymization for each figure, respectively. We can find some interesting patterns from the visualization. For example, the length of the neck bone is slightly increased, and the bone is moved to the upright position after anonymization. Given that an elderly female acts, we can conjecture that the adjustment makes gender unrecognizable. More visualization results are provided in Appendix E.

## 4 Related work

### 4.1 Public dataset anonymization

Researchers have pointed out privacy issues with public visual datasets and tried to mitigate them. Caesar et al. (2020); Frome et al. (2009); Yang et al. (2021) propose a blurring approach where the privacy-sensitive regions are blurred with an object detection method. Flores and Belongie (2010); Uittenbogaard et al. (2019) propose an inpainting method to remove potentially problematic objects such as pedestrians and vehicles. A large body of prior work has used GANs (Goodfellow et al., 2014) to preserve visual private information. Ren et al. (2018); Maximov et al. (2020) use GANs to generate fake faces to replace real ones.

There are also some works that exist for other domains: sound domain (Cohen-Hadria et al., 2019; Sümer et al., 2020) and text domain (Li et al., 2018; Coavoux et al., 2018; Mosallanezhad et al., 2019). Similar concerns are also made for skeleton datasets. Sinha et al. (2013) propose a method to recognize persons from skeleton data. This work focuses on gait patterns extracted from human skeletons. This implies the potential privacy leakage from public datasets.

### 4.2 Skeleton-based action recognition

Human skeleton data is a sequence of graphs, where joints and bones are represented as nodes and edges separately within a graph. In early times, skeleton motion trajectories are embedded into a manifold space as points. The relative distances between these points acted as clues for action recognition. However, these models do not exploit the internal spatial relationship between joints. Later, convolution neural networks (CNNs) are utilized to extract spatial co-occurrence patterns between joints. Nevertheless, CNNs cannot model a skeleton's topological information.

Then, graph convolution networks (GCNs) are introduced to model these topological relations. Nonetheless, basic GCNs are not suitable for human skeleton sequences because they contain not only the 3D position of joints but also the time series. Yan et al. (2018) introduce the spatial-temporal graph convolutional networks (ST-GCN). They conduct graph convolution for extracting spatial features and perform $1 \times 1$ convolution over each joint for capturing temporal variations. Following this line, various graph neural architectures are proposed to extract features from the graphs: AS-GCN (Li et al., 2019), AGC-LSTM (Si et al., 2019), 2s-AGCN (Shi et al., 2019), MS-G3D (Liu et al., 2020) and Shift-GCN (Cheng et al., 2020).

In this work, we use Shift-GCN(Cheng et al., 2020), MS-G3D(Liu et al., 2020), and 2s-AGCN(Shi et al., 2019) as a baseline recognition model for private information. Although the original model is developed to recognize the actions of skeletons, we empirically show the model can successfully classify the private information with a proper training procedure.

## 5 Conclusion

In this work, we investigate privacy leakage from publicly available skeleton datasets. We show that although skeleton data may seemingly be privacy-protective, recently proposed skeletal action

recognizers are surprisingly capable of extracting sensitive and identity information from these data. To address this privacy leakage problem, we propose a learning framework. Our experimental results reveal that the proposed method effectively removes the privacy information while preserving the movement patterns. Note that the anonymizers used in this work employ relatively simple architectures. Experiments show that private information can be removed effectively even with simple architectures. We leave the study of more advanced architectures for future work since our goal is to show the potential vulnerability of the skeletons and to provide a general framework to overcome.

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

# A Detailed Experimental Setting

In this section, we describe the detailed experimental setting such as anonymization networks and datasets. Additionally, the code used in our experiments is available at `https://anonymous.4open.science/r/Skeleton-anonymization-90B7/`. Since the implementation is similar for all baselines, we only provide the code based on Shift-GCN. Also, data related to ETRI-activity3D will not be provided, since ETRI-activity3D can access by authorized users. If one wants to use this dataset for research purposes, please visit here[2] and get permission. For all experiments, we set $k = 1$ (see in Algorithm 1), i.e., one minimization with one maximization. We use four NVIDIA GeForce RTX 3090 or four NVIDIA RTX A5000 for anonymization training.

## A.1 Anonymization Networks

**Residual network.** The residual network (He et al., 2016) anonymizer adopts a simple residual connection from the input skeletons to the output skeletons. Specifically, the model can be formalized as

$$f_\theta(\vec{x}) = \text{MLP}_\theta(\vec{x}) + \vec{x},$$

where $\text{MLP}_\theta : \mathbb{R}^{D \times 3} \to \mathbb{R}^{D \times 3}$ is a simple multi-layered perceptron parameterized by $\theta$. The residual connection keeps the position similar to the original skeleton while the MLP layer models the disposition of joints to anonymize. We use two fully-connected layers to model the disposition. The anonymizer is applied to each frame of a skeleton sequence. Although the anonymizer is applied to each frame independently, the back-propagated signals from action and private classifiers make the entire sequence coherent. By initializing $\theta$ with weights close to zero, we make the anonymizer add a small random noise to the original skeleton in the early stage of learning.

**U-Net.** The U-Net (Ronneberger et al., 2015) architecture is adopted to our anonymizer network. The U-Net consists of two paths: the contracting path, and the expanding path. In the contracting path, it repeats downsampling and maxpool an input skeleton data to encode it to the feature map. In the expanding path, U-Net repeats upsampling and concatenating feature map via skip connection. Especially, skip connections concatenates the features from the contracting path to the corresponding level in the expanding path. It makes the output skeleton position similar to the original skeleton. We use U-Net architecture from Pytorch-UNet repository[3] for our anonymizer network.

## A.2 Datasets

**ETRI-activity3D.** ETRI-activity3D is an action recognition dataset originally published for recognizing the daily activities of the elderly and youths. It contains 112,620 skeleton sequence samples observed from 100 people, half of whom were between the ages of 64 and 88 and the rest were in their 20s. The elderly consist of 33 females and 17 males, and the young adults consist of 25 females and 25 males. The samples are categorized into 55 classes based on the activity type. Each action is captured from 8 different Kinect v2 sensors to provide multiple views. Each sequence consists of 3D locations of 25 joints of the human body.

With the ETRI-activity3D dataset, we anonymize the gender information from the skeletons. We drop samples from 5 classes for the following experiments, e.g., handshaking, containing two people, so only one person appears in the remaining samples. After removing malformed and two-person samples, we split the remaining samples into 68,788 and 34,025 training and validation, respectively. We split the dataset according to the subject ID. In other words, the subjects in the validation set do not appear in the training set. Through this split, we measure the generalizability of the gender classifier to the unknown subjects.

**NTU60.** NTU60 is an action recognition dataset that contains 60 action classes and 56,880 skeleton sequences taken from 40 subjects. The format of skeleton data is the same as ETRI-activity3D, which includes the 3D positions of 25 human body joints. After removing malformed samples, we split the remaining samples into 37,646 and 18,932 as training and validation sets, respectively. Following

---

[2]https://ai4robot.github.io/etri-activity3d-en/
[3]https://github.com/milesial/Pytorch-UNet

the original work (Shahroudy et al., 2016), we split it according to the camera ID so that both sets contain identical subjects with different views.

### A.3 Detailed Algorithm of Adversarial Anonymization

Algorithm 1 is algorithm that we explained in

---

**Algorithm 1** Adversarial Anonymization

---

**Require:** Pre-trained classifiers $h_\psi$ and $g_\phi$, $E$: # of epochs, $m$: minibatch size, $k$: # of minimization steps
  **while** until convergence **do**
    **for** $t \leftarrow 1$ to $k$ **do**
      Sample minibatch of $m$ samples $\{(\vec{x}_i, y_i, z_i)\}_{i=1}^m$
      Compute $\nabla_\theta \mathcal{L}_{\text{adv}}$ with minibatch             ▷ Equation 2
      Update $\theta \leftarrow \theta - \nabla_\theta \mathcal{L}_{\text{adv}}$
    **end for**
    Sample minibatch of $m$ samples $\{(\vec{x}_i, y_i, z_i)\}_{i=1}^m$
    Compute $\nabla_\phi \alpha \, \text{CE}\left(z, g_\phi\left(f_\theta(\vec{x})\right)\right)$ with minibatch
    Update $\phi \leftarrow \phi - \nabla_\phi \alpha \, \text{CE}\left(z, g_\phi\left(f_\theta(\vec{x})\right)\right)$
  **end while**

---

## B Privacy Leakage

Table 2 and Table 3 provide the detailed results of privacy leakage experiments. We use three baseline models: Shift-GCN[4], MS-G3D[5], and 2s-AGCN[6]. In this experiment, we set all hyperparameter configurations as default values given baseline models. To obtain a stable result, multiple training with random initialization is conducted with two NVIDIA GeForce RTX 3090 or two NVIDIA RTX A5000.

Table 2: Top-1 and Top-5 accuracy of re-identification task with NTU60 dataset. The re-identification task achieves 80% and 97% for top-1 and top-5 accuracy on average.

|  | Top-1 | $\sigma^2$ | Top-5 | $\sigma^2$ |
|---|---|---|---|---|
| Shift-GCN | 0.7962 | 0.0070 | 0.9681 | 0.0009 |
| MS-G3D | 0.8223 | 0.0087 | 0.9751 | 0.0007 |
| 2s-AGCN | 0.7689 | 0.0183 | 0.9656 | 0.0032 |

Table 3: Accuracy of gender classification with ETRI-activity3D dataset. The gender classifier achieves 87% accuracy on average.

|  | Accuracy | $\sigma^2$ |
|---|---|---|
| Shift-GCN | 0.8599 | 0.0040 |
| MS-G3D | 0.8790 | 0.0017 |
| 2s-AGCN | 0.8643 | 0.0047 |

## C Additional Analysis

### C.1 Validation Accuracy Analysis

We plot how the validation accuracy changes over the training procedure with adversarial learning in Figure 5. For both datasets, the action accuracy remains high over the epochs on both training

---

[4]https://github.com/kchengiva/Shift-GCN
[5]https://github.com/kenziyuliu/MS-G3D
[6]https://github.com/lshiwjx/2s-AGCN

and validation sets. However, there is a gap between the training and validation accuracy on privacy, which shows the overfitting in the classifier.

Specifically, for gender classification, the gender accuracy starts similar value. Then gender accuracy increases on the train set and decreases on the validation set. The re-identification accuracy drops first on both training and validation sets for the re-identification task. Note that the re-identification task achieves 80% accuracy. The validation accuracy at the first epoch indicates that the re-identification task is more sensitive to the additional noise introduced by random weights of the residual network than the gender classification.

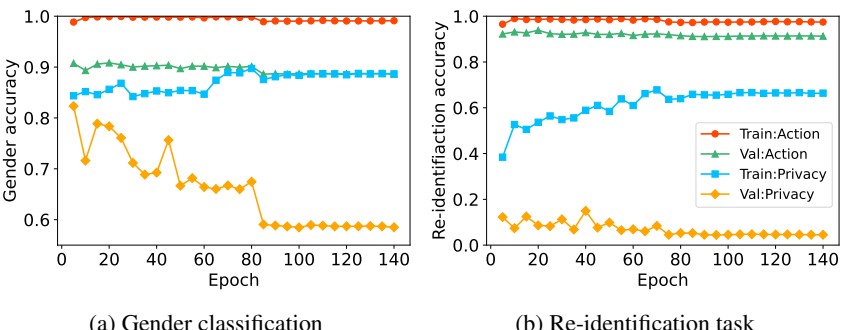

(a) Gender classification                 (b) Re-identification task

Figure 5: Accuracy and reconstruction error over epochs with the residual anonymizer. 'Train:Action' and 'Val:Action' indicate the training and validation accuracy of the action classification, and 'Train:Privacy' and 'Val:Privacy' indicate the training and validation accuracy of the privacy classification.

## C.2    Reconstruction Error Analysis

A reconstruction error directly shows the difference between the original and anonymized skeletons. Although we cannot directly set the level of reconstruction error, we vary the parameters to obtain different levels of reconstruction error and corresponding prediction accuracy. Please check the below for the detailed hyperparameter settings. As shown in Figure 6, there is a trade-off between reconstruction error and re-identification accuracy. As we increase the reconstruction error, we can reduce the re-identification accuracy. However, high reconstruction error yields low action accuracy as well.

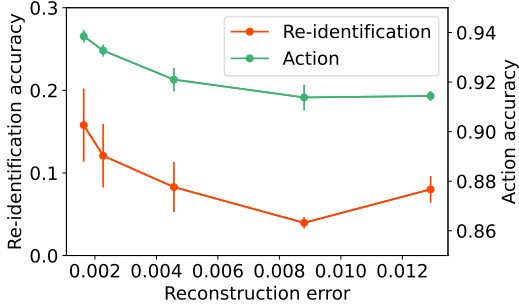

Figure 6: The trade-off between reconstruction error and accuracy with the residual anonymizer on NTU60.

## C.3    Hyperparameter analysis

In this subsection, we analyze the hyperparameter used in our objective function (See Equation 2) with action accuracy, re-identification accuracy, and reconstruction error. There are two hyperparameters: $\alpha$ and $\beta$. $\alpha$ adjusts the importance of privacy, one expects that large $\alpha$ may occur low privacy accuracy. Also, $\beta$ adjusts the reconstruction error, which regularizes anonymized data to look similar to original data. We conduct this experiment with Shift-GCN, NTU60, and ResNet.

**Alpha term analysis.**   Figure 7 is the result of different $\alpha$. In this experiment, we vary $\alpha$ with fixed $\beta$ of 10. As shown in Figure 7a, action accuracy decreases, as $\alpha$ increases. Also, re-identification accuracy drops rapidly from 0.1 to 1.0, but increases at 2.0 and decreases again. Note that $\alpha$: 1, $\beta$: 10 is the best model in our metric. This means that optimizing the minimax objective works well under a certain $\alpha$ so that growing $\alpha$ produces low re-identification accuracy. However, $\alpha$ that exceeds a particular value disturbs optimization due to a trade-off with action accuracy and reconstruction error. Also, Figure 7b shows that reconstruction error increases, as $\alpha$ increases. Since the ratio of $\beta$ to $\alpha$ decreases, so regularizing anonymized data can not influence enough during training.

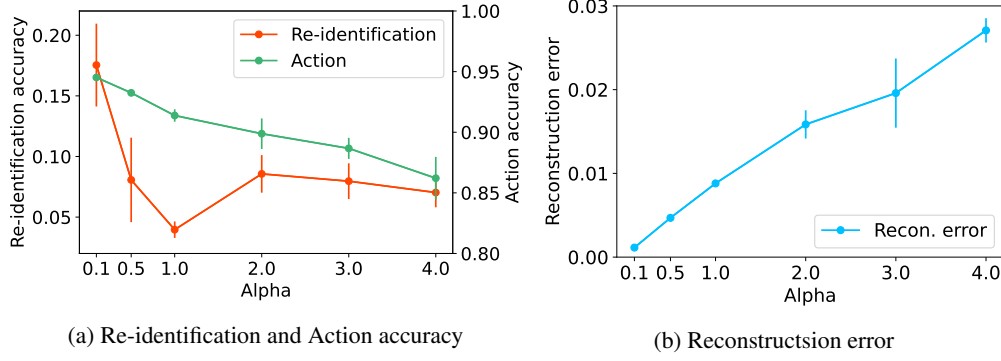

(a) Re-identification and Action accuracy

(b) Reconstructsion error

Figure 7: Re-identification accuracy, action accuracy, and reconstruction error with different $\alpha$

**Beta term analysis.**   Figure 8 is the result of according to different $\beta$. In this experiment, we vary $\beta$ with fixed $\alpha$ of 1. As shown in Figure 8b, it is trivial that reconstruction error increases by raising $\beta$. This means that large $\beta$ makes anonymized data look similar to original data. This affects re-identification accuracy and action accuracy. Figure 8a shows that action accuracy and re-identification accuracy increase, as $\beta$ increases.

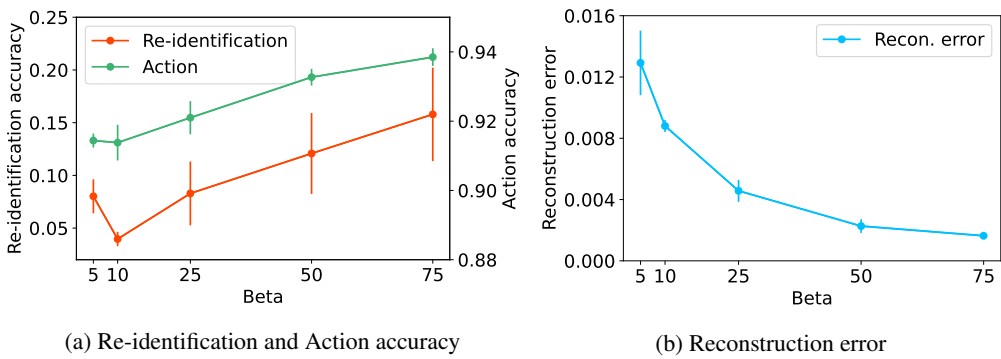

(a) Re-identification and Action accuracy

(b) Reconstruction error

Figure 8: Re-identification accuracy, action accuracy, and reconstruction error with different $\beta$

# D   Detailed Experimental Results

In this section, detailed hyperparameter configurations and results of our main experiments are provided.

**Anonymization results.**   Table 4 and Table 5 provide the detailed hyperparameter configurations and the entire results of Figure 2. For all experiments, we train Shift-GCN to find a representative model at first. After finding the best $\alpha$ and $\beta$, the same $\alpha$ and $\beta$ are applied to the other two baselines with different learning rates. As the result, we set $\alpha$ and $\beta$ as 1.0 and 10 at ResNet, 0.3 and 2 at U-Net for NTU60. Also, we set $\alpha$ and $\beta$ as 2.0 and 15 at ResNet, 0.3 and 0.5 at U-Net for ETRI-activity3D, respectively.

Table 4: Detail hyperparameter configurations and results of our representative model with three baselines for anonymizing identity information using NTU60.

| Model | $lr$ | Re-iden. | Action. | Recon. error |
|---|---|---|---|---|
| Shift-Res | 0.0100 | 0.04202 | 0.9175 | 0.00860 |
| Shift-U | 0.0100 | 0.05701 | 0.9145 | 0.01655 |
| MS-Res | 0.0005 | 0.07770 | 0.9215 | 0.01092 |
| MS-U | 0.0005 | 0.10110 | 0.9243 | 0.01115 |
| 2s-Res | 0.0050 | 0.05889 | 0.8994 | 0.01446 |
| 2s-U | 0.0050 | 0.03514 | 0.8908 | 0.07536 |

Table 5: Detail hyperparameter configurations and results of our representative model with three baselines for anonymizing gender information using ETRI-activity3D.

| Model | $lr$ | Gender. | Action. | Recon. error |
|---|---|---|---|---|
| Shift-Res | 0.010 | 0.5875 | 0.8863 | 0.00127 |
| Shift-U | 0.010 | 0.5671 | 0.8708 | 0.01086 |
| MS-Res | 0.001 | 0.6426 | 0.9073 | 0.00108 |
| MS-U | 0.001 | 0.6046 | 0.9033 | 0.00403 |
| 2s-Res | 0.005 | 0.4221 | 0.8758 | 0.00397 |
| 2s-U | 0.005 | 0.5645 | 0.8563 | 0.00920 |

**Trade-off analysis results.** Table 7 and Table 8 show the detailed results of Figure 3. We vary hyperparameter $\alpha$ and $\beta$ to observe the trade-off between action accuracy and privacy accuracy. For these experiments, we use Shift-GCN and set the learning rate as 0.01.

**Reconstruction error analysis results.** Table 6 provides the detailed results of Figure 6. To obtain different levels of reconstruction error, we vary $\beta$ with fixed $\alpha$ as 1. For these experiments, we use Shift-GCN with NTU60 and set the learning rate as 0.01. Multiple training with random initialization is conducted for stable results.

Table 6: Detail results of reconstruction error analysis.

| $\beta$ | Recon. error | Action | | Re-iden. | |
|---|---|---|---|---|---|
| | | Acc. | $\sigma^2$ | Acc. | $\sigma^2$ |
| 5 | 0.012920 | 0.9144 | 0.00203 | 0.08016 | 0.01612 |
| 10 | 0.008804 | 0.9138 | 0.00515 | 0.03964 | 0.00680 |
| 25 | 0.004570 | 0.9210 | 0.00478 | 0.08288 | 0.03034 |
| 50 | 0.002268 | 0.9327 | 0.00241 | 0.12080 | 0.03837 |
| 75 | 0.001636 | 0.9385 | 0.00255 | 0.15790 | 0.04433 |

Table 7: Results with varying hyperparameters for analyzing the trade-off between action accuracy and re-identification accuracy with NTU60.

| $\alpha$ | $\beta$ | Action acc. | Re-iden. acc. | Recon. error (RMSE) | Anonymization Network |
|---|---|---|---|---|---|
| 1 | 5 | 0.9191 | 0.08303 | 0.00980 | ResNet |
| 1 | 10 | 0.9175 | 0.04202 | 0.00860 | ResNet |
| 1 | 15 | 0.9351 | 0.17070 | 0.00380 | ResNet |
| 1 | 20 | 0.9235 | 0.12010 | 0.00550 | ResNet |
| 2 | 5 | 0.8909 | 0.13190 | 0.02431 | ResNet |
| 2 | 10 | 0.8838 | 0.09381 | 0.01822 | ResNet |
| 2 | 15 | 0.9113 | 0.14690 | 0.01008 | ResNet |
| 2 | 20 | 0.8728 | 0.07691 | 0.01020 | ResNet |
| 3 | 5 | 0.8731 | 0.08266 | 0.02557 | ResNet |
| 3 | 10 | 0.8814 | 0.07500 | 0.01340 | ResNet |
| 3 | 15 | 0.8891 | 0.09286 | 0.01649 | ResNet |
| 3 | 20 | 0.8595 | 0.10550 | 0.01410 | ResNet |
| 0.3 | 0.5 | 0.9188 | 0.08657 | 0.02463 | U-Net |
| 0.3 | 1 | 0.9206 | 0.07622 | 0.01945 | U-Net |
| 0.3 | 1.5 | 0.9209 | 0.07891 | 0.01513 | U-Net |
| 0.3 | 2 | 0.9145 | 0.05701 | 0.01655 | U-Net |
| 0.5 | 0.5 | 0.9060 | 0.08045 | 0.02959 | U-Net |
| 0.5 | 1 | 0.9074 | 0.06956 | 0.06496 | U-Net |
| 0.5 | 1.5 | 0.9099 | 0.07791 | 0.02055 | U-Net |
| 0.5 | 2 | 0.9071 | 0.05509 | 0.01750 | U-Net |
| 0.7 | 0.5 | 0.8945 | 0.06597 | 0.12950 | U-Net |
| 0.7 | 1 | 0.9022 | 0.05277 | 0.07635 | U-Net |
| 0.7 | 1.5 | 0.8988 | 0.08494 | 0.02410 | U-Net |
| 0.7 | 2 | 0.9006 | 0.04352 | 0.02232 | U-Net |

Table 8: Results with varying hyperparameters for analyzing the trade-off between action accuracy and gender accuracy with ETRI-activity3D.

| $\alpha$ | $\beta$ | Action acc. | Gender acc. | Recon. error (RMSE) | Anonymization Network |
|---|---|---|---|---|---|
| 1 | 5 | 0.9028 | 0.6648 | 0.0017160 | ResNet |
| 1 | 10 | 0.9032 | 0.6778 | 0.0007986 | ResNet |
| 1 | 15 | 0.9050 | 0.7643 | 0.0004499 | ResNet |
| 1 | 20 | 0.9032 | 0.6655 | 0.0003264 | ResNet |
| 2 | 5 | 0.8940 | 0.6569 | 0.0061400 | ResNet |
| 2 | 10 | 0.8925 | 0.6296 | 0.0023480 | ResNet |
| 2 | 15 | 0.8863 | 0.5875 | 0.0012730 | ResNet |
| 2 | 20 | 0.9011 | 0.7352 | 0.0011160 | ResNet |
| 3 | 5 | 0.8890 | 0.6612 | 0.0061500 | ResNet |
| 3 | 10 | 0.8934 | 0.6512 | 0.0035020 | ResNet |
| 3 | 15 | 0.9031 | 0.7601 | 0.0025420 | ResNet |
| 3 | 20 | 0.8958 | 0.6558 | 0.0020570 | ResNet |
| 0.3 | 0.5 | 0.8708 | 0.5671 | 0.0108600 | U-Net |
| 0.3 | 1 | 0.8754 | 0.5798 | 0.0094480 | U-Net |
| 0.3 | 1.5 | 0.8337 | 0.5892 | 0.0264000 | U-Net |
| 0.3 | 2 | 0.8758 | 0.5952 | 0.0108300 | U-Net |
| 0.5 | 0.5 | 0.8569 | 0.5797 | 0.0156400 | U-Net |
| 0.5 | 1 | 0.8303 | 0.5905 | 0.0276500 | U-Net |
| 0.5 | 1.5 | 0.8341 | 0.5853 | 0.0249800 | U-Net |
| 0.5 | 2 | 0.8567 | 0.5823 | 0.0117300 | U-Net |
| 0.7 | 0.5 | 0.8239 | 0.6053 | 0.0294500 | U-Net |
| 0.7 | 1 | 0.8265 | 0.5978 | 0.0326100 | U-Net |
| 0.7 | 1.5 | 0.8269 | 0.5984 | 0.0277300 | U-Net |
| 0.7 | 2 | 0.7504 | 0.6046 | 0.0469700 | U-Net |

# E    Qualitative Analysis

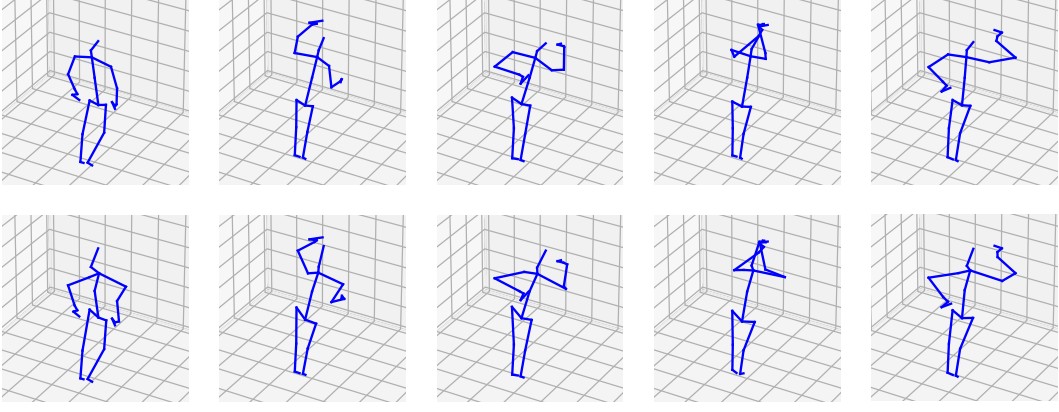

(a) The original (top) and the gender anonymized (bottom) skeletons for an action "wiping face with a towel" from ETRI-activity3D. The subject is an elderly female.

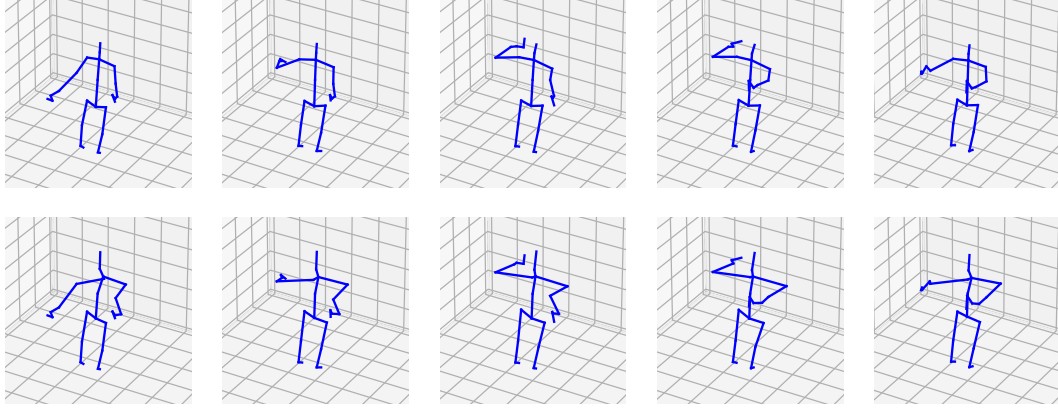

(b) The original (top) and the gender anonymized (bottom) skeletons for an action "drinking water" from ETRI-activity3D. The subject is an elderly male.

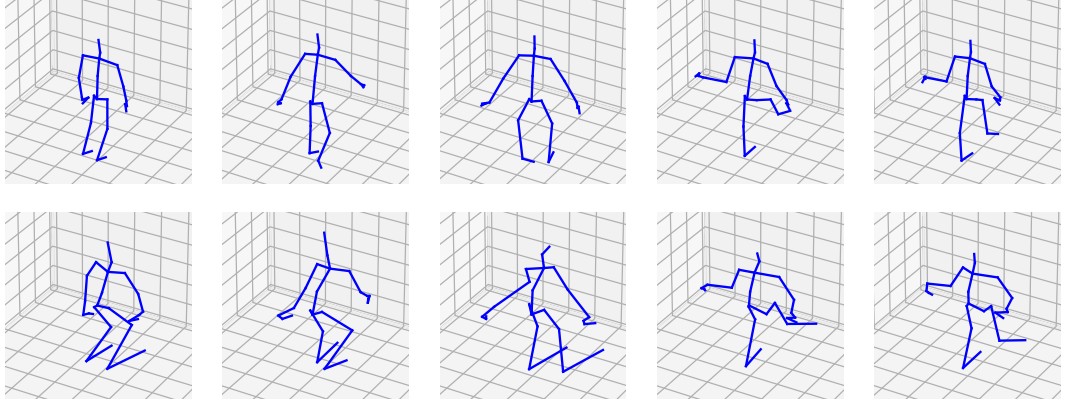

(c) The original (top) and the identity anonymized (bottom) skeletons for an action "kicking something" from NTU60. The subject's ID is 2.

Figure 9: Examples of anonymized skeletons. Five frames are visualized from a sequence of action frames.

