# OpenReview forum: "Anonymization for Skeleton Action Recognition"
_NeurIPS.cc/2022/Workshop/TSRML — TSRML2022_

### Official Review · Reviewer_2Mc8 · 2022-10-11
**Good paper about privacy preserving.**

**Overall Rating:** 7

**Summary:**

The authors propose a framework to anonymize based on adversarial learning to protect potential privacy leakage (gender) from the skeleton dataset.




**Strengths:**

 - First approach in this direction.
 - Evaluation: Different datasets are used.
 - While privacy is protected. Action recognition is only marginal hampered.


**Weaknesses:**

 - Skeleton recognition is very often used in videos. It is not clear if this framework hampers the speed of the recognition.
 - E Qualitative Analysis: c) second row. The feet do not show real. It looks modified.
 - Can it be extended to more characteristics as only gender or even combined?

Writing:
 - page 10. A3 - No introduction and description of Algorithm 1. Same for E Qualitative Analysis.
 - Figure 7 shows action accuracy. Why is not there any graphic which splits up the actions?
 - Line 174: linespace missing after GANs.



**Overall Recommendation:**

This paper shows an approach for anonymization for skeleton action recognition. The author could prove that gender anonymization is possible without significantly degrading the action recognition. Some skeleton modifications are visible to the human eye.


**Review Confidence:**

3: The reviewer is fairly confident that the evaluation is correct

---

### Official Review · Reviewer_EqND · 2022-10-18
**A meaningful advancement for an important problem**

**Overall Rating:** 6

**Summary:**

The authors study the problem of data anonymization for skeleton action recognition datasets. They suggest a way to remove identifiable or sensitive information from the dataset to guarantee the safety of personal data. Although skeleton data format seems to be more anonymized than the RGB-images, the authors demonstrate that the personal information of the subjects can be extracted from there as well. They propose to train an anonymizer network to prevent potential privacy leakage. The effect of this dataset modification on model training performance is demonstrated to be marginal since the information required for the proper action recognition usually remains intact after anonymization.

**Strengths:**

The paper studies an interesting problem of data anonymization. The authors propose a novel framework and a way to evaluate the level of privacy in the data using privacy classifier.

The paper is well-organized and easy to follow. It provides numerous illustrations that support the discussion.

The performance of the re-identification models is shown to be hindered by anonymization, therefore, demonstrating the efficiency of the approach.

**Weaknesses:**

The paper has raised awareness of the problem of privacy leakage in the skeleton action recognition datasets by considering the leakage of gender and id (from a given set of ids). For the future work it would be interesting to see the performance of the method for other possible personal information such as age, physical characteristics etc.

**Overall Recommendation:**

The paper addresses an important problem of privacy leakage and provides a novel and sound solution of training an anonymization network. The discussion is well structured and the empirical results are sufficient, therefore I recommend the acceptance.

**Review Confidence:**

3: The reviewer is fairly confident that the evaluation is correct

---

### Official Review · Reviewer_7GLs · 2022-10-23
**Interesting problem**

**Overall Rating:** 6

**Summary:**

This paper focuses on anonymization of skeletons for action recognition, and the privacy leakage of action recognition models. The proposed method in this paper is an adversarial minimax game, where the utility of the action recognition model is maximized, as the power of an adversary to predict private attribute is minimized.

**Strengths:**

The paper claims to be the first studying anonymization and leakage of action recognition models

**Weaknesses:**


1. The paper has taken an already existing method which is commonly used (adversarial learning for privacy) [1-3] and applied it to a new domain, which is not really novel. However, the experimentations are thorough.

2. It's better if gender is not selected as a sensitive attribute,   since it's not really binary and classifying it based on given skeleton features seems insensitive.

[1] Mireshghallah F, Taram M, Jalali A, Elthakeb AT, Tullsen D, Esmaeilzadeh H. Not all features are equal: Discovering essential features for preserving prediction privacy. InProceedings of the Web Conference 2021 2021 Apr 19 (pp. 669-680).

[2] Ossia, Seyed Ali et al. “Deep Private-Feature Extraction.” *IEEE Transactions on Knowledge and Data Engineering*
 32 (2020): 54-66.

[3] Huang C, Kairouz P, Chen X, Sankar L, Rajagopal R. Generative adversarial privacy. arXiv preprint arXiv:1807.05306. 2018 Jul 13.

**Overall Recommendation:**

Although the paper's approach isn't completely new, the problem is interesting and studying it is important

**Review Confidence:**

4: The reviewer is confident but not absolutely certain that the evaluation is correct

---

### Decision · Program_Chairs · 2022-10-23

Accept